# Efficiency in Focus: LayerNorm as a Catalyst for Fine-tuning Medical Visual Language Pre-trained Models

## ABSTRACT

In the realm of Medical Visual Language Models (Med-VLMs), the quest for universal efficient fine-tuning mechanisms remains paramount, especially given researchers in interdisciplinary fields are often extremely short of training resources, yet largely unexplored. Given the unique challenges in the medical domain, such as limited data scope and significant domain-specific requirements, evaluating and adapting Parameter-Efficient Fine-Tuning (PEFT) methods specifically for Med-VLMs is essential. Most of the current PEFT methods on Med-VLMs have yet to be comprehensively investigated but mainly focus on adding some components to the model's structure or input. However, fine-tuning intrinsic model components often yields better generality and consistency, and its impact on the ultimate performance of Med-VLMs has been widely overlooked and remains understudied. In this paper, we endeavour to explore an alternative to traditional PEFT methods, especially the impact of fine-tuning Layer Normalization (LayerNorm) layers, Feedforward Neural Networks and Attention layers on the Med-VLMs. Our comprehensive study spans both small-scale and large-scale Med-VLMs, evaluating their performance under various fine-tuning paradigms across tasks such as Medical Visual Question Answering and Medical Imaging Report Generation. The findings reveal unique insights into the effects of intrinsic parameter fine-tuning methods on fine-tuning Med-VLMs to downstream tasks and expose fine-tuning solely the LayerNorm layers not only surpasses the efficiency of traditional PEFT methods but also retains the model's accuracy and generalization capabilities across a spectrum of medical downstream tasks. The experiments show LayerNorm fine-tuning's superior adaptability and scalability, particularly in the context of large-scale Med-VLMs. We hope this work will contribute to the ongoing discourse on optimizing efficient fine-tuning strategies for Med-VLMs. The code will be released upon acceptance.

## CCS CONCEPTS

• **Computing methodologies** → **Information extraction**; *Neural networks*; • **Information systems** → *Multimedia streaming*.

## KEYWORDS

Medical Visual Language Models, Efficient Fine-tuning, Layer Normalization

**Unpublished working draft. Not for distribution.**

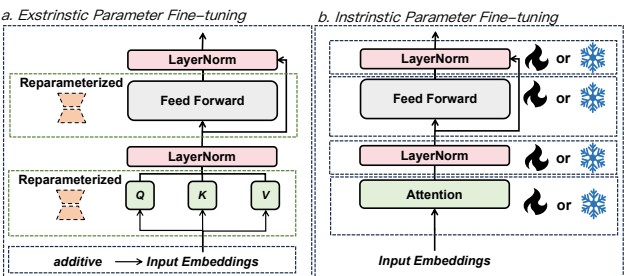

**Figure 1: Illustration of the (a)extrinsic parameter fine-tuning and (b)intrinsic parameter fine-tuning methods.**

## 1 INTRODUCTION

Visual language models (VLMs) have become pivotal in facilitating multimodal tasks within the medical domain, such as medical visual question answering (Med-VQA) and medical imaging report generation (Med-IRG). The pretraining-finetuning paradigm, heralded for its success in domain adaptation and transfer, now stands as the predominant training approach for Medical VLMs (Med-VLMs). Nonetheless, the substantial data and computational resource demands for VLM pretraining pose significant challenges. Despite the success of visual language pre-training paradigms like CLIP [27] and BLIP [18] fostering a series of open-source medical visual language pre-trained (VLP) models contributed by the community, adapting these models for specific downstream tasks remains a formidable task for those constrained by resource availability. Especially considering the inherent minor variability in medical imaging across different medical centers or imaging devices, which necessitates researchers to frequently and rapidly fine-tune medical VLP models (Med-VLPs).

The recent surge in Large Visual Language Models (LVLMs) has exacerbated these challenges. Although a series of Parameter-Efficient Fine-Tuning (PEFT) methods[11, 12, 20, 25] have been developed in the Large Language Model (LLM) domain, their applicability and effectiveness in the context of LVLMs are yet to be ascertained[2]. Some empirical studies[29] have shown that the performance of certain PEFT methods contradicts their demonstrated competitiveness in the original LLM domain when fine-tuning domain-specific LVLMs because fine-tuning parameters for different parts of the LVLM can have drastically different effects. Distinct from universal domains, the medical field presents unique challenges, such as limited dataset size and specialized data characteristics, which necessitate a tailored approach to fine-tuning. These domain-specific requirements underscore the need for a dedicated evaluation of PEFT methods on Med-VLMs to ensure their efficacy and appropriateness for medical tasks. Even so, research on the impact of imposing different PEFT methods on different scale Med-VLMs for different tasks remains completely blank. Meanwhile, current PEFT methods typically focus on domain adaptation

by adding extra components to the model structure (*i.e.,* Reparameterized Fine-tuning) [12] or prefixes to the input (*i.e.,* Additive Fine-tuning) [10, 20, 25], while the role of fine-tuning intrinsic structural parameters of models has been widely neglected, especially in vertical domain fine-tuning. As shown in Figure1

In this paper, we focus on efficiently fine-tuning Med-VLP models for specific downstream tasks, aiming to explore an innovative method that achieves task adaptation by fine-tuning a small subset of the model's intrinsic parameters. To find a universally efficient fine-tuning method applicable to various types of Med-VLMs, regardless of their scale, we turn our attention to common foundational layers in transformer-based components, including attention layers, Feed-forward Neural Networks (FFN), and Layer Normalization (LayerNorm) layers. Through systematic experiments, we demonstrate that LayerNorm can serve as the most efficient method for fine-tuning Med-VLPs. To further explore the efficiency, adaptability, and performance of fine-tuning the LayerNorm layer for transferring Med-VLPs to downstream tasks, We have conducted an extensive evaluation across both large-scale and small-scale Med-VLPs, encompassing core medical visual language tasks like Med-VQA and Med-IRG. Our findings expose the variability of intrinsic parameter fine-tuning methods in fine-tuning Med-VLMs to a downstream task that is different from traditional extrinsic parameter fine-tuning methods. We not only underscore the superior efficiency of LayerNorm fine-tuning over existing PEFT methods but also its remarkable adaptability and transferability for fine-tuning Med-VLPs across diverse downstream tasks. The main contributions of this article are as follows:

- To our best knowledge, we are the first to centre on fine-tuning a small subset of the Med-VLP's inherent parameters to adapt to downstream tasks.
- We conduct a comprehensive series of experiments fine-tuning foundational components of Med-VLMs, including systematic comparisons with existing PEFT methods centred on tuning extrinsic components.
- Our research identifies LayerNorm fine-tuning as a highly efficient strategy for adapting Med-VLPs to specific downstream tasks within the medical domain.

## 2 RELATED WORK

### 2.1 Medical Visual Language Models

In the medical domain, Med-VLMs play a pivotal role in automating visual-language tasks, such as VQA and IRG. Initially, these models [7, 9, 21, 26] leverage Convolutional Neural Networks (CNNs) and Recurrent Neural Networks (RNNs) to extract visual and linguistic features separately. Yet, such approaches frequently fell short in terms of generalizability and transferability across different tasks due to the limitations of their structure. Modern Med-VLMs [2–4] primarily adopt the transformer architecture, following a pretraining-finetuning paradigm. They undergo initial pretraining on extensive, generalized medical image-text pair datasets, followed by comprehensive fine-tuning on more focused, task-specific datasets. For example, MISS [3], utilizing the ALBEF [19] methodology, begins its training on 38,800 selectively curated image-text pairs from the MedICaT dataset before undergoing fine-tuning for

VQA tasks. Similarly, LLaVA-Med [16] employs a dual-phase pretraining strategy, starting with image-text feature alignment on two million pairs from PubMed, then enhancing conversational capabilities using instruction-format data, culminating in full-scale fine-tuning for VQA tasks. These approaches consistently rely on full-model fine-tuning for task adaptation, a method that, despite its efficacy, demands substantial resources, particularly for large-scale models such as LLaVA-Med. The restricted dataset sizes available for downstream task training further jeopardize the model's generalizability, leading to potential catastrophic forgetting and diminishing its broader applicability in medical contexts.

### 2.2 Efficient Fine-tuning Techniques

The fine-tuning of large-scale Pre-trained Language Models (PLMs) is a demanding process [1, 32, 34], requiring extensive computational resources and data. To alleviate these burdens, PEFT techniques [10, 20, 25, 29, 31], have been introduced. These methods [11, 12, 24] typically incorporate trainable components into the PLMs while maintaining the rest of the model's parameters in a frozen state. Some strategies [15, 20, 25] also involve the nuanced manipulation of input embeddings across different layers to minimize or negate modifications to the original model's architecture.

PEFT methods have demonstrated efficacy in transitioning large-scale PLMs to new tasks or downstream applications and have been instrumental in converting LLMs into multimodal LLMs [5, 17, 32, 34]. For instance, LLaVA [23] uses an MLP adapter to connect a vision feature extractor with a large language model, selectively training the MLP adapter while keeping both components static, thus adapting the LLM into a VLM. [33] introduces an efficient strategy where tuning LayerNorm layers suffices to yield strong performance to transform an LLM into an LVLM. Nonetheless, the capability of existing PEFT methods to efficiently adapt pre-trained VLMs to specialized, especially medical, tasks remains largely uninvestigated. With the diverse architectures of LVLMs, the most effective application of PEFT methods is uncertain, and their generalizability to non-textual encoders/decoders is limited (*e.g.,* prefix-tuning and p-tuning are not viable for Vision Transformers (ViT) [8]). Consequently, investigating the adjustment of a model's intrinsic parameters for efficient fine-tuning emerges as a critical necessity. In this paper, we propose a novel method that eschews adding components to the original model structure or input, focusing instead on fine-tuning the model's inherent parameters. This strategy is designed to ensure the method's broad applicability for efficient fine-tuning across various Med-VLM types.

## 3 PRELIMINARIES

### 3.1 Mainstream Architectures of Med-VLMs

Contemporary generative Med-VLMs, irrespective of their scale—be it large-scale or small-scale, tend to follow a similar architectural framework. This typical structure comprises a vision feature extractor, a text feature extractor, a connector that integrates the former two, and a Language Model (LM) head. Most Med-VLMs opt for ViT [8] as the vision feature extractor, while the text encoder is based on mainstream frameworks such as BERT [6] or GPT [1]. Despite possible minor variations in their structural implementations, the transformer-based layer serves as their common

 

denominator, with FFN, Attention mechanisms, and LayerNorm being indispensable core components.

## 3.2 Previous PEFT Methods

Transitioning from the core mechanisms of attention and layer normalization, which provide stability and specificity within the model's architecture, we delve into the domain of extrinsic PEFT methods. These methods are categorized primarily into two types: Reparameterized Fine-tuning (*i.e.,* LoRA (Low-Rank Adaptation) and Additive Fine-tuning (*i.e.,* Prefix-tuning).

**LoRA-Tuning:** LLM maps data into a high-dimensional space for processing. LoRA indirectly trains the dense layers in the network by optimizing the rank-decomposition matrix that changes in the adaptation process of the dense layer, thereby achieving the best fine-tuning effect by optimizing only the rank-decomposition matrix of the dense layer. For the pretrained parameters $\theta_0^D$, the dense layer weight parameter matrix on a specific downstream task is defined as $W_0 \in \mathbb{R}^{d \times k}$ and the intrinsic rank of it is $\theta^d$; the specific downstream task's parameters $\theta^D$ is calculated as $\theta^D = \theta_0^D + \theta^d M$, where $M$ is the rank-decomposition matrix. For $W_0 \in \mathbb{R}^{d \times k}$, LoRA updates it with the following equation:

$$W_0 + \Delta W = W_0 + BA, B \in \mathbb{R}^{d \times r}, A \in \mathbb{R}^{r \times k}, \quad (1)$$

where $d$ is the output dimension of the previous layer, and $k$ is the input dimension of the next layer. For input $x$, the forward propagation process is calculated as follows:

$$h = W_0 x + \Delta W x = W_0 x + BA x. \quad (2)$$

**Prefix-Tuning:** Inspired by the In-Context Prompting method adopted by GPT3 [1], Li *et al* [20] propose the Prefix-tuning method for generation tasks. Instead of the discrete text used in prompt tuning, continuous vectors are prefixed to the input text. Specifically, the generation task is deemed as a table-to-text task, the input $x$ is treated as a linear table and the output $y$ represents a short text. For an encoder-decoder model, different prefixes are attached to the beginning of the encoder and decoder with the input defined as: $z = [PREFIX, x, PREFIX']$, and the prefixes are generated by a trainable matrix $P_\theta \in \mathbb{R}^{|P_{idx}| \times dim(h_i)}$, the global training objective is defined as:

$$\max_\phi \log P_\phi(y|x) = \max_\phi \sum_{i \in Y_{idx}} \log P_\phi(z_i|h_{<i}). \quad (3)$$

## 3.3 Medical Visual Language Tasks

**Medical Visual Language Answering:** The primary objective of Med-VQA is to provide answers based on professional questions posed by the inquirer regarding medical images, enhancing the understanding of medical images, and facilitating patient care through the automated interpretation of visual data. The tasks are categorized into open-ended questions, which require detailed descriptive answers, and close-ended questions, which demand concise, often binary responses like "yes" or "no". This interdisciplinary domain requires the Med-VLM to interpret and provide insights into complex medical imagery, such as X-rays, MRI scans, and CT images.

**Medical Imaging Report Generation:** Medical IRG involves the automatic creation of textual descriptions for medical images, using

Med-VLMs. This task aims to analyze visual medical data and produce accurate, coherent, and clinically relevant reports. The goal is to assist radiologists and healthcare professionals by reducing their workload and improving diagnostic efficiency while maintaining high standards of patient care.

## 3.4 Difference between Universal VLMs and Med-VLMs' Downstream Tuning

The differences between Med-VLMs and Universal VLMs make it necessary to study tuning of Med-VLMs separately in addition to the PEFT approach for Universal VLMs.

From the data perspective, the dataset used for downstream task fine-tuning in the medical domain is extremely narrow compared to the universal domain, for example, the current largest radiological image dataset used for the Med-VQA task includes only 14,028 image-question pairs, which makes the fine-tuning of LVLMs fall into the problems of overfitting and catastrophic forgetting. At the same time, the answers of the textual Instruction pairs embedded in the dataset usually include only one or two simple words. Furthermore, the current training loss used by generative models in the fine-tuning process makes it very easy to fall into the learning of the data distribution from the long text to its short text rather than the learning of the correct image-text association. From the model perspective, most Med-VLMs are obtained by transfer learning from VLMs in the universal domain, a process where the visual coder is usually frozen, however, the domain gap between natural images and medical images affects the performance of Med-VLMs on medical tasks, where most of the current PEFT methods are not available for ViTs or or have not demonstrated their effective impact on visual encoders for transfer learning from general domain to medical domain. Therefore, a separate study of efficient fine-tuning methods for Med-VLM on downstream tasks is necessary.

## 4 TUNING SETTINGS

### 4.1 Baseline Model

To explore a method that achieves task adaptation by fine-tuning a small subset of the model's own parameters, we choose two different-scale pre-trained Med-VLMs for different tasks: 1) small-scale VLM MISS [3] and 2) large-scale VLM LLaVA-Med [16] for Med-VQA and Med-IRG. These baseline models cover generative Med-VLM at different scales and for different tasks so that we can provide comprehensive insights into the impact of different PEFT methods on fine-tuning Med-VLM to downstream tasks.

### 4.2 Tuning within Transformer-based Layer

The transformer-based layer, serving as a fundamental structure across vision encoders, language models, and certain VLM connectors, undergoes fine-tuning through both intrinsic parameter adjustments and the application of extrinsic component fine-tuning methods. This process aims to assess the impact of various tuning approaches on the overall model performance.

**Strategic Intrinsic Adjustments:** We emphasize the transformer-based layer's role as the computational core of the model and selectively fine-tune its intrinsic parameters. Attention layers, LayerNorm layers, and FFNs constitute the critical units of this layer. As

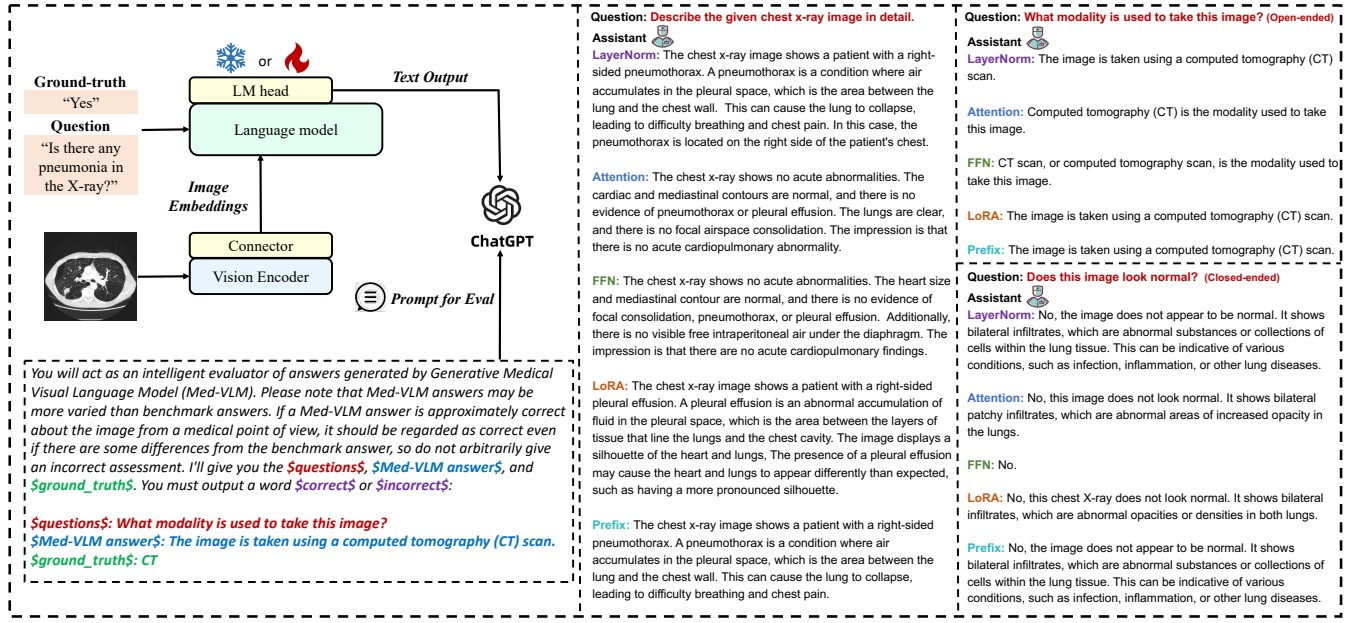

**Figure 2: The pipeline of our study. The flowchart details the step-by-step process from input reception to output generation, showcasing the model's method for processing medical images and questions to generate contextually relevant responses. The right side presents the fine-tuning results across different paradigms, including both Med-VQA and Med-IRG tasks.**

depicted in Figure1, when one of these three components is set to be trainable, the remaining parameters within the transformer-based layer are kept frozen.

**Incorporation of PEFT Techniques:** LoRA-tuning and Prefix-tuning, representing the forefront of Parameter-Efficient Fine-Tuning (PEFT) methods, are chosen for comparison against intrinsic parameter adjustments. Figure1 illustrates the application details of these PEFT methods within our study. For LoRA-tuning, low-rank matrices are selectively applied to the parameters of the query and value matrices within the attention layer, mapping data into a low-dimensional subspace for processing. In the case of Prefix-tuning, we follow prevalent practices by appending prefix vectors to the origin input $x$ of the key and value within the attention layer, and the final input embeddings can be defined as $z = [PREFIX, x]$.

## 4.3 Tuning within the VLM Architecture

In our pursuit to uncover the nuanced impact of various modules within the full VLM's architecture on overall model performance, we embark on a strategic fine-tuning expedition which entails selectively training specific modules within the VLM framework while employing efficient fine-tuning methods for certain components or maintaining others in a frozen trainable state. Such a strategy allows us to dissect the individual contributions of each component to the model's efficacy in medical visual language tasks, offering insights into optimizing Med-VLMs for enhanced performance and efficiency. This selective fine-tuning approach aims to validate the hypothesis that certain components within the Med-VLM architecture wield more significant influence over the model's performance on medical tasks. By applying focused fine-tuning strategies to

individual modules, we seek to delineate the performance impact of targeted adjustments versus broad model updates.

**Details for Small-scale Med-VLMs:** For small-scale Med-VLMs, such as MISS, we experiment with applying efficient fine-tuning techniques to some modules at a time, with the rest of the model's parameters set to remain fully trainable This is because for small-scale med-VLM, making either module completely frozen may make the model unable to transfer to downstream tasks [2]. For instance, when the language model undergoes LayerNorm-tuning(LN-tuning), the vision feature extractor, connector, and LM head are kept in a state that allows full parameter adjustments. This strategy allows us to compare the impact of different efficient fine-tuning methods on model performance under the premise that we can evaluate the impact of fine-tuning different module parameters of small-scale Med-VLMs on the overall task performance.

**Strategy for Large-scale Med-VLMs:** In the case of large-scale Med-VLMs, like LLaVA-Med, our fine-tuning strategy is more nuanced, reflecting the diverse requirements of comparative analysis. As shown in Figure 2, 'Snowflakes or flames' indicate that any module has the option to be adjusted or frozen. Depending on the specific experimental setup, modules within these larger models may be categorized into three states: fully trainable (T), efficiently fine-tuned some of the parameters (PEFT), and completely frozen (F). This flexible approach comprehensively evaluates how different tuning states across various modules influence large-scale VLMs' performance on complex medical visual language tasks.

## 4.4 Downstream Fine-tuning

**Benchmarks:** In this paper, we use a total of four datasets, Slake Dataset [22] and VQA-RAD dataset [14] for Med-VQA, and OpenI dataset [30] and MIMIC dataset[13] for Med-IRG. The Slake dataset consists of 14,028 QA pairs, of which 70% are used for training, 15% for validation, and 15% for testing. The VQA-RAD dataset is used for the zero-shot performance of the model on the VQA task including 3515 QA pairs, of which 451 pairs are used for testing. The OpenI dataset [30] is used for the training of the Med-IRG task including 6,459 images and 3,955 reports, the instructions are the same as those adopted in [28]. The MIMIC test set [13] was chosen for the evaluation of the model's Med-IRG performance, which includes 5,159 images and 3,269 reports. For the inference of the IRG task, we uniformly use the phrase "Describe the given chest x-ray image in detail." as the instruction.

**Application Details:** For the fine-tuning of our chosen models (MISS and LLaVA-Med), a consistent set of hyperparameters is employed to ensure uniformity across our experiments. Each model is fine-tuned with an initial learning rate of 2e-5, utilizing the Adam optimizer for its well-regarded efficiency in handling the optimization landscapes of deep learning models. Specifically, MISS underwent training for 120 epochs with a batch size of 16, adopting a weight decay of 0.05 to encourage regularization. In contrast, LLaVA-Med's fine-tuning is characterized by a warmup ratio of 0.03 and a cosine learning rate scheduler, alongside specific adjustments such as enabling tensor float 32 for enhanced computational performance, and employing FSDP strategies for memory efficiency, with settings like "full_shard auto_wra" and targeting the "LlamaDecoderLayer" for wrapping. During all the inferences, Med-LLaVA generates outputs using a set of predefined generation parameters, including sampling methods and beam search configurations (numbeams=1), and the temperature is kept at 0.2. Detailed information regarding the hyperparameter settings for each model, along with additional configurations and the rationale for their selection, is provided in the *Appendix* for further reference.

## 5 EXPERIMENT RESULTS AND DISCUSSION

### 5.1 Small-scale MISS Result

As shown in Table 1, we employ both supervised fine-tuning (SFT) and performance assessment on the Slake dataset's training and testing sets. The reported performance metrics include accuracy rates for 'opened' and 'closed' types, which means open-ended and closed-ended questions, as well as a global accuracy rate that averages the performance across both types.

Given the potentially catastrophic impact of freezing any module on the overall performance of small-scale models, when certain modules underwent efficient fine-tuning, the remaining modules were maintained fully trainable. In the context of the MISS model, ViT, JTM, and DEC represent the visual encoder, joint text-multimodal encoder, and text decoder, respectively. The term "trainable params" refers to the total volume of trainable model parameters, with "#Params" indicating the ratio of trainable to total parameters. "PEFT params" denotes the proportion of parameters fine-tuned using PEFT methods, with "#PEFT Params" reflecting the proportion of PEFT-tuned parameters relative to the total parameters within the corresponding module.

When the baseline model is fully fine-tuned, it achieves the highest open-ended question accuracy and global accuracy rates of 82.91% and 82%, respectively. Under the premise of maintaining the visual encoder fully trainable and only efficiently tuning one module at a time, keeping the JTM encoder fully trainable enabled the model to achieve optimal performance. Compared to scenarios where DEC underwent full parameter training while JTM was efficiently tuned, the model's global accuracy rates under LayerNorm, attention, and FFN intrinsic parameter tuning methods were higher by 4%, 9%, and 12%, respectively. However, the outcomes with LoRA-tuning and Prefix-tuning are inconsistent with intrinsic parameter tuning methods, attributed to the total volume of parameters trained under these two scenarios being inversely related.

Maintaining the visual encoder fully trainable while efficiently tuning all the remaining modules resulted in significantly poor model performance, failing to correctly judge the closed-source questions in all the fine-tuning methods except attention-tuning. Under LayerNorm, FFN, LoRA, and Prefix-tuning methods, the model never answered 'yes' to any close-ended question, with accuracy rates lower than random guessing at 38.03%, 54.65%, 50.70%, and 32.95%, respectively.

Comparing different fine-tuning methods, the effect of LN-tuning is remarkable, achieving the best accuracy on close-ended questions with the lowest PEFT Params, even surpassing full parameter tuning and reaching 84.51%. In contrast, although attention-tuning and FFN-tuning slightly outperform LN-tuning in terms of global accuracy, this came at the cost of tuning over 40% of the parameters in their respective modules. The LoRA method fine-tuning model using the [T, PEFT, T] paradigm tuned approximately five times more PEFT parameters than LN-tuning (only 56,823), with Prefix-tuning at twenty times more. This underscores the viability of LN-tuning as a comparable method to the most classical PEFT methods in small-scale fine-tuning scenarios that require saving certain parameter volumes. From a global parameter tuning perspective, attention-tuning achieved performance closest to full fine-tuning by saving 23% of trainable parameters, marking it as another viable fine-tuning approach for small-scale Med-VLMs.

### 5.2 Large-scale LLaVA-Med Result

Furthermore, we conduct comprehensive evaluations on LLaVA-Med, a large-scale model designated for Med-VQA tasks. Our approach encompassed three distinct training paradigms: [PEFT, F, PEFT, F], [PEFT, T, PEFT, T], [F, F, PEFT, F], and [F, T, PEFT, T]. Considering the substantial parameter size of LVLMs, we aimed to restrict the volume of fine-tuning parameters to within about 40%, thereby excluding full parameter training of the ViT and FFN-tuning methods that involve adjusting ViT. Table 2 showcases the experimental results of LLaVA-Med, trained and tested on the Slake dataset, employing the aforementioned fine-tuning paradigms.

When opting to keep both the connector and LM head trainable, the model's performance did not exhibit significant improvement, despite a substantial increase in the volume of adjusted parameters. Specifically, when fine-tuning adopted the [F, T, PEFT, T] paradigm, changes in global accuracy rates for LN, attention, and FFN tuning compared to [F, F, PEFT, F] are -0.5%, -2.1%, and -3.3% respectively.

**Table 1: Comparison of accuracy (ACC-%) of MISS on Slake dataset using different methods of fine-tuning. 'T' stands for trainable while 'F' stands for frozen.**

| ViT | JTM | DEC | Opened ↑ | Closed ↑ | Gobal ↑ | Trainable Params | #Params | PEFT Params | #PEFT Params |
|---|---|---|---|---|---|---|---|---|---|
| T | T | T | **82.91** | 81.47 | **82.00** | 361,478,972 | 100% | - | - |
| T | LayerNorm | LayerNorm | 40.79 | 38.03 | 39.87 | 86,454,528 | 23.92% | 115,200 | 0.04% |
| T | LayerNorm | T | 75.64 | **84.51** | 78.61 | 224,277,308 | 62.04% | **56,832** | **0.04%** |
| T | T | LayerNorm | 73.65 | 77.46 | 74.93 | 223,656,192 | 61.87% | 58,368 | 0.04% |
| T | Attention | Attention | 64.51 | 74.65 | 71.25 | 199,806,720 | 55.27% | 113,467,392 | 41.24% |
| T | Attention | T | 78.47 | 85.92 | 80.96 | 280,954,172 | **77.72%** | 56,733,696 | 41.33% |
| T | T | Attention | 75.5 | 64.23 | 71.72 | 280,331,520 | 77.55% | 56,733,696 | 41.15% |
| T | FFN | FFN | 74.79 | 54.65 | 68.05 | 199,677,696 | 55.24% | 113,338,368 | 41.19% |
| T | FFN | T | 76.63 | **84.51** | 79.26 | 280,889,660 | 77.71% | 56669184 | 41.27% |
| T | T | FFN | 76.20 | 49.86 | 67.39 | 280,267,008 | 77.53% | 56669184 | 41.10% |
| T | LoRA | LoRA | 68.14 | 50.70 | 62.29 | 86,929,152 | 24.05% | 589,824 | 0.21% |
| T | LoRA | T | 76.77 | 82.81 | 78.79 | 224,515,388 | 62.11% | 294,912 | 0.21% |
| T | T | LoRA | 78.52 | 79.44 | 78.83 | 223,892,736 | 61.94% | 294,912 | 0.21% |
| T | Prefix | Prefix | 41.50 | 32.95 | 38.61 | 115,884,288 | 32.06% | 29,544,960 | 10.74% |
| T | Prefix | T | 75.92 | 83.38 | 78.42 | 238,992,956 | 66.12% | 14,772,480 | 10.76% |
| T | T | Prefix | 76.82 | 82.25 | 78.65 | 238,370,304 | 65.94% | 14,772,480 | 10.71% |

**Table 2: Comparison of SOTA methods adopting pre-training and fine-tuning paradigm but with different numbers of pre-trained images on open-ended accuracy (ACC).**

| Vision Tower | Connector | LLM | LM head | Slake Dataset | | | #Params | Trainable Params | BERTS-Recall | Mean Token |
|---|---|---|---|---|---|---|---|---|---|---|
| | | | | Opened ↑ | Closed ↑ | Global ↑ | | | | |
| LayerNorm | F | LayerNorm | F | 59.53 | **69.95** | 63.62 | 3.79% | 266,737,664 | 46.35% | **28.27** |
| F | F | LayerNorm | F | 58.76 | 69.71 | 63.05 | 0.00372% | **262,144** | 46.24% | 27.81 |
| LayerNorm | T | LayerNorm | T | 59.84 | 67.55 | 62.87 | 3.79% | 266,737,664 | 46.93% | 26.81 |
| F | T | LayerNorm | T | 60.31 | 66.11 | 62.58 | 3.78% | 266,637,312 | 46.93% | 26.50329877 |
| Attention | F | Attention | F | 61.4 | 67.79 | 63.9 | 31.91% | 2,248,245,248 | 49.25% | 25.246 |
| F | F | Attention | F | **61.71** | 68.03 | 64.18 | 30.48% | 2,147,483,648 | 49.11% | 25.95 |
| Attention | T | Attention | T | 60.93 | 65.87 | 62.87 | 35.69% | 2,514,620,416 | 48.47% | 25.89066918 |
| F | T | Attention | T | 58.76 | 66.83 | 61.92 | 34.26% | 2,413,858,816 | 48.49% | 25.8539114 |
| F | T | FFN | T | 64.5 | 62.26 | 63.62 | 44.74% | 3,152,056,320 | 51.98% | 16.42318567 |
| F | F | FFN | F | **64.34** | 66.59 | **65.22** | 40.96% | 2,885,943,296 | 52.07% | 17.37983035 |
| F | F | LoRA | F | 58.14 | 64.42 | 60.6 | 0.14% | 9,994,240 | 47.55% | 25.43 |
| F | T | LoRA | T | 58.76 | 65.38 | 61.36 | 3.92% | 276,369,408 | 47.26% | 25.72196041 |
| F | F | Prefix | F | 56.9 | 67.07 | 60.89 | 15.48% | 1,090,805,760 | 46.19% | 26.61639962 |
| F | T | Prefix | T | 59.22 | **70.19** | 63.52 | 19.26% | 1,357,180,928 | 46.28% | 26.58906692 |

This contradicts the common notion that more parameter adjustments correlate with better SFT performance, indicating that full parameter adjustments of the connector and LM head during efficient fine-tuning of LLMs do not guarantee the expected outcomes.

The performance changes are inconsistent under the [PEFT, F, PEFT, F] and [PEFT, T, PEFT, T]. For LayerNorm (LN)-tuning, fine-tuning the image encoder led to respective increases in global accuracy of 0.57% and 0.29%, while attention-tuning resulted in changes of -0.28% and +0.85%. Such subtle differences do not conclusively indicate whether adjusting parameters of the image encoder benefits or hinder model performance, especially when considering Recall metrics. The increase in ViT-adjusted parameter volume did not regularly alter recall, suggesting that larger adjustments to ViT parameters do not consistently improve model recall.

Comparing different intrinsic parameter adjustments revealed that increasing the volume of fine-tuned parameters indeed enhances the model's recall of generated content: as fine-tuning parameters shifted from 0.003% to 44%, recall correspondingly increased from 46.24% to 52.07%. This indicates that enlarging the volume of fine-tuned parameters allows the model to learn the

distribution of ground-truth tokens in the vocabulary space more effectively, both quantitatively and spatially. However, considering accuracy—a gold standard in medical tasks—significant increases in parameter volume do not necessarily elevate all accuracy metrics concurrently. LN-tuning under the [PEFT, F, PEFT, F] paradigm once again achieved state-of-the-art (SOTA) accuracy for close-ended questions, which was consistent with observations in small-scale VLMs. Across two models of different scales, pre-trained on distinct datasets and tasks, LN-based fine-tuning consistently enhanced their accuracy on close-ended questions.

While attention-tuning and FFN-tuning marginally surpassed LN-tuning in global accuracy, achieving peak open-ended question accuracies of 64.34% and global accuracy of 65.22%, this came at the cost of escalating the volume of tuned parameters from 262,144 to 2,885,943,296—a millionfold increase. Furthermore, following peak performance under current fine-tuning paradigms, the model ceased learning intrinsic relations of the features, instead focusing on the quantitative distribution of ground-truth tokens. This shift manifested in minimal accuracy improvements and a dramatic reduction in average output length, with mean tokens dropping from

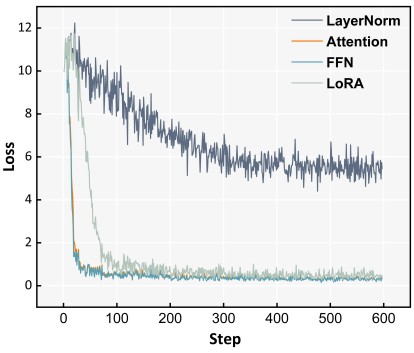

**Figure 3: Loss curves for various methods under the [F, T, PEFT, T] fine-tuning paradigm.**

28.27 to 17.38. Figure 2 compares the generated outcomes across different intrinsic and extrinsic fine-tuning methods under the [F, F, PEFT, F] paradigms, illustrating this phenomenon driven by inherent large model training patterns and suboptimal training data. Most datasets employed for fine-tuning Med-VQA tasks comprise answers in short text formats, with close-ended answers typically being 'yes' or 'no', and open-ended answers containing only a few words. For LLMs seeking interpretability of the answer, adjusting more parameters paradoxically impairs generative performance.

When applying LoRA-tuning and Prefix-tuning to LLaVA-Med, the model's performance did not exhibit notable improvement. LoRA-tuning's recall reached 47.55% and 47.26%, indicating a closer alignment of the model's output with the ground truth distribution in the vocabulary space. However, this did not translate to enhanced evaluation accuracy, with accuracies under [F, F, PEFT, F] and [F, T, PEFT, T] fine-tuning paradigms reaching only 60.6% and 61.36%, respectively. This suggests that LoRA-tuning failed to deepen the multimodal model's understanding of joint image-text features, merely aligning output closer to the ground truth distribution.

Figure 3 displays the loss curves for various methods under the [F, T, PEFT, T] fine-tuning paradigm. Compared to LN-tuning, LoRA-tuning's minimum fine-tuned parameter volume is approximately fifty times larger, yet its accuracy is roughly 3% lower, with average output lengths of 25.43 and 25.72, trailing behind LN-tuning. These factors collectively indicate that existing PEFT methods may not directly enhance text-based accuracy in discerning medical images in multimodal model downstream fine-tuning, underscoring the advantages of LN-tuning over traditional PEFT approaches.

### 5.3 Large-scale VLM IRG Result
To further explore the impact of various fine-tuning methods on the performance of large-scale Med-VLMs in the Medical Imaging Report Generation (Med-IRG) context, we employed the [F, F, PEFT, F] fine-tuning paradigm using the OpenI dataset to fine-tune Med-LLaVA. The model's OOD performance was then tested on the MIMIC dataset's test set to assess how it handles variations in input text domains. As shown in Table3, the performance of the models fine-tuned with LN, Attention, LoRA, and Prefix methods showed minimal differences, with output text lengths averaging around 122.7. In contrast, FFN tuning significantly outperformed other

**Table 3: Comparison of LLaVA-Med competence on the MIMIC test dataset which SFT on the OpenI dataset.**

| Vision Tower | Connector | LLM | LM head | METEOR score | Rouge-L | | | Mean Token |
|---|---|---|---|---|---|---|---|---|
| | | | | | Recall | Precision | F1 | |
| F | F | LayerNorm | F | 12.85% | 12.58% | 15.92% | 13.61% | 122.66 |
| F | F | Attention | F | 12.85% | 12.58% | 15.92% | 13.61% | 122.66 |
| F | F | FFN | F | **24.53%** | **17.01%** | **23.84%** | **19.34%** | **123.11** |
| F | F | LoRA | F | 12.95% | 12.57% | 15.93% | 13.62% | 122.70 |
| F | F | Prefix | F | 12.99% | 12.47% | 15.92% | 13.54% | 122.72 |

fine-tuning approaches, demonstrating its superior capability in learning the underlying representations of ground-truth in long text generation tasks like Med-IRG.

### 5.4 Out of Distribution Performance Testing
To assess whether the performance of LLaVA-Med on a familiar dataset like Slake correlates with its performance on a novel dataset, we conducted an OOD testing on the VQA-RAD dataset. This test serves to evaluate the model's robustness and flexibility by applying it to a different domain within the same field but with unseen data. More specifically, the images in the VQA-RAD dataset belong to the proximity domain with the Slake dataset but are quite different from the Slake dataset in terms of question formulation. Such experiments allow us to consider the ability of different fine-tuning methods to reason on non-proximity-domain text over similar medical images, in order to speculate on the ability of the models fine-tuned with the VQA dataset to be applied to the real Med-VQA scenarios.

In this experiment, we observe various fine-tuning paradigms, focusing particularly on the role of the transformer-based Layer-Norm, Attention, and FFN adjustments. The results show a notable variance in the model's ability to generalize the learned features to the VQA-RAD dataset. As Table4 shows, fine-tuning methods that show comparable results on the Slake dataset exhibit significant performance variances on the OOD VQA-RAD dataset. Notably, under the [F, T, PEFT, T] training paradigm, LoRA-tuning underwent a remarkable reversal, surpassing the performances of attention-tuning and FFN-tuning, which were previously effective on the Slake dataset. It achieves a global accuracy of 65.41% and matches the best closed question accuracy of 73.71%, initially noted with LN-tuning. Conversely, FFN-tuning, despite being superior at learning adjacent training text representations, disappointed in its OOD performance. While it excelled in Rouge-L metrics with scores of 26.35%, 9.80%, and 13.51%, significantly surpassing other fine-tuning methods, it only managed accuracy scores of 56.50%, 64.54%, and 60.98%. Moreover, its mean output length plummeted to 18.51, the lowest among all methods. This combination of metrics further validates that although FFN-tuning can closely fit the training data during SFT, it predominantly learns the distribution of ground-truth tokens rather than enhancing the model's ability to generalize image-text reasoning. A similar pattern is observed with attention-tuning, the global accuracy decreases by 7.54% compared to LN-tuning when the connector and LM head are trained more, however, the text length is optimal at this time. Comprehensively, the mean token length of attention tuning under the same paradigm in Table 1 can show that attention tuning slightly overlearns the text in the adjacent domains during SFT training, but does not cause

**Table 4: Comparison of LLaVA-Med competence on the VQA-RAD test dataset which SFT on the Slake dataset.**

| Vision Tower | Connector | LLM | LM head | VQA-RAD | | | Bertscore | | | METEOR Score | Rouge-L | | | Mean Token |
|---|---|---|---|---|---|---|---|---|---|---|---|---|---|---|
| | | | | Opened↑ | Closed↑ | Global↑ | Precision | Recall | F1 | | Recall | Precision | F1 | |
| F | T | LayerNorm | T | 54.50 | **73.71** | 65.19 | 29.77% | 49.00% | 36.53% | 12.53% | 7.91% | 1.48% | 2.30% | 29.52 |
| F | T | Attention | T | 50.50 | 63.35 | 57.65 | 29.97% | 49.27% | 36.73% | 11.87% | 7.97% | 1.46% | 2.30% | 31.12 |
| F | T | FFN | T | 56.50 | 64.54 | 60.98 | 35.44% | 52.05% | 41.89% | 19.23% | 26.35% | 9.80% | 13.51% | 18.50 |
| F | T | LoRA | T | 55.00 | **73.71** | **65.41** | 30.06% | 49.52% | 36.87% | 12.03% | 7.89% | 1.50% | 2.29% | 27.52 |
| F | T | Prefix | T | 51.00 | 70.12 | 61.64 | 29.89% | 48.85% | 36.58% | 12.71% | 8.53% | 1.57% | 2.45% | 28.87 |
| F | F | LayerNorm | F | 54.50 | **75.30** | **66.08** | 29.90% | 48.84% | 36.60% | 12.97% | 7.84% | 1.50% | 2.33% | 29.52 |
| F | F | Attention | F | 55.50 | 71.71 | 64.52 | 30.25% | 49.64% | 37.08% | 8.44% | 8.44% | 1.66% | 2.58% | 29.45 |
| F | F | FFN | F | 52.00 | 61.35 | 57.21 | 35.20% | 51.76% | 41.60% | 18.43% | 24.31% | 8.96% | 12.40% | 19.16 |
| F | F | LoRA | F | 49.50 | 70.92 | 61.42 | 30.08% | 49.80% | 36.96% | 12.46% | 7.91% | 1.51% | 2.33% | 29.04 |
| F | F | Prefix | F | 51.00 | 69.72 | 61.42 | 29.76% | 48.49% | 36.36% | 12.81% | 8.11% | 1.54% | 2.41% | 29.21 |

**Table 5: Comparison of LLaVA-Med zero-shot competence on the MIMIC test dataset which SFT on the Slake dataset.**

| Vision Tower | Connector | LLM | LM head | METEOR score | Rouge-L | | | Mean Token |
|---|---|---|---|---|---|---|---|---|
| | | | | | Recall | Precision | F1 | |
| F | T | LayerNorm | T | 11.86% | 11.27% | 18.54% | 13.54% | 71.78 |
| F | T | Attention | T | 11.88% | 11.15% | 17.98% | 13.33% | 73.73 |
| F | T | FFN | T | 12.12% | 11.24% | 18.07% | 13.42% | 73.23 |
| F | T | LoRA | T | 11.57% | 10.99% | 17.91% | 13.18% | 71.51 |
| F | F | LayerNorm | F | 12.07% | 11.40% | 18.26% | 13.56% | 75.00 |
| F | F | Attention | F | 12.17% | 11.50% | 18.20% | 13.65% | 75.01 |
| F | F | FFN | F | 12.81% | 11.48% | 17.84% | 13.53% | 76.88 |
| F | F | LoRA | F | 11.91% | 11.25% | 18.07% | 13.40% | 72.93 |

large damage to the model's output ability of textual diversity in textual reasoning in the non-adjacent domains.

When enlarging the perspective to compare the accuracy performances across different fine-tuning methods, LN-tuning consistently displayed formidable strength. Under the [F, F, PEFT, F] tuning paradigm, LN-tuning, utilizing the smallest parameter adjustment, reached the highest scores in opened, closed and global accuracy—54.5%, 75.3%, and 66.08%, respectively. It also maintained a longer mean text output than any other methods under the same tuning conditions. These results, coupled with the Rouge-L metrics from Table1, indicate that LN-tuning manages to enhance the model's understanding of multimodal feature interrelations, significantly minimizing the model's overemphasis on learning ground-truth text token distributions due to low training data quality. This is evidenced by the lowest recall rate of 1.5% and the highest global accuracy of 66.08%. In contrast, LoRA-tuning, despite adjusting 50 times more parameters, did not significantly outperform LN-tuning.

## 5.5 Zero-shot Capability Investigation

To further investigate the zero-shot capabilities of different fine-tuning paradigms on Med-VLMs, we conduct extensive evaluations of the LLaVA-Med model on the MIMIC test dataset after SFT on the Slake dataset. This analysis aims to understand the impact of various intrinsic tuning methods on the model's ability to generalize and adapt to new tasks within the medical domain, particularly for IRG tasks. The evaluation employs metrics such as METEOR score, Rouge-L, and mean token length to measure factual accuracy, linguistic precision, and diversity of output in medical report generation. From Table 5, experimental results indicate that different tuning methods exhibit varying impacts on the model's zero-shot performance. LN-tuning consistently showed robust performance across different configurations, achieving the highest precision

scores (18.54% and 18.26% under different paradigms), which underscores its effectiveness in preserving the factualness of model outputs. In contrast, the Attention and FFN methods, although effective in some scenarios, demonstrated greater variability in their influence on model generalization.

Notably, FFN-tuning, which previously excelled in VQA tasks, scored the lowest in precision (17.84%) under the [F, F, FFN, F] paradigm on the MIMIC dataset. This suggests that the model may have overlearned task-specific features from the Slake dataset, thus hindering its generalization and transfer capabilities. Furthermore, METEOR scores positively correlated with the number of adjusted parameters, increasing from 11.86% to a high of 12.81%, indicating that a larger volume of tuned parameters enhances the model's linguistic alignment capabilities in medical text generation tasks. Examining the effects of freezing versus tuning the connector and LM head reveals no clear pattern in performance metrics between [F, T, PEFT, T] and [F, F, PEFT, F] configurations. Changes in Rouge-L scores and precision were minimal, suggesting that extensive fine-tuning of the connector and LM head does not necessarily contribute to improved zero-shot performance across these metrics.

Comparison between intrinsic tuning and traditional methods such as LoRA-tuning did not exhibit standout performance in the zero-shot setting. Under the [F, T, PEFT, T] paradigm, LoRA-tuning shows lower METEOR scores, Rouge-L, and mean token length compared to intrinsic methods, indicating that LoRA-tuning might not effectively maintain the overall transferability and generalization of the model in medical applications. Thus, intrinsic tuning methods, particularly LN-tuning with minimal parameter adjustments, might be a better choice, especially under the [F, F, PEFT, F] paradigm, where it outperforms more parameter-intensive methods like attention-tuning in maintaining the generalization capabilities. These observations underscore the efficacy of LN-tuning in preserving the generalization of LVLMs for diverse medical tasks.

## 6 CONCLUSION

This study presents a thorough examination of intrinsic parameter fine-tuning and exposing LN-tuning, as a potent alternative to traditional PEFT methods for Med-VLMs. Our extensive experimental analysis across both small-scale and large-scale Med-VLMs demonstrated that fine-tuning the LayerNorm layers significantly enhances the models' adaptability, efficiency, and scalability in performing specialized medical tasks, such as Med-VQA and Med-IRG. We hope this work will enhance the clinical applicability of Med-VLMs in real-world medical settings.

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
