# OpenReview forum: "Efficiency in Focus: LayerNorm as a Catalyst for Fine-tuning Medical Visual Language Models"
_acmmm.org/ACMMM/2024/Conference — MM2024 Poster_

### Official Review · Reviewer_eioU · 2024-05-24

**Rating:** 4
**Confidence:** 3

**Summary:**

This paper starts discussing the current Parameter-Efficient Fine-Tuning (PEFT) methods on Med-VLMs and then explore an alternative to traditional PEFT methods by fine-tuning a small subset of the Med-VLP’s inherent parameters. Lsatly, the paper identifies LayerNorm fine-tuning as a highly efficient strategy for adapting Med-VLPs to specific down stream tasks within the medical domain.

**Strengths:**

+ The paper reveal unique insights into the effects of intrinsic parameter fine-tuning methods on fine tuning Med-VLMs to downstream tasks and expose fine-tuning solely the LayerNorm layers not only surpasses the efficiency of traditional PEFT methods but also retains the model’s accuracy and generalization capabilities across a spectrum of medical downstream tasks.
+ This paper conducted extensive fine-tuning experiments on two large language models in the medical field, targeting multiple downstream medical imaging tasks.
+ The experiments are comprehensive, and the results are convincing, leading to valuable conclusions.
+ This research is significant for empowering large language models in the medical domain.

**Limitations:**

- Can this method be tested on more medical imaging tasks to demonstrate its effectiveness?
- To my knowledge, there are existing fine-tuning methods beyond LoRA, such as P-tuning and P-tuning v2. How does your method compare to these fine-tuning methods? Can the experimental conclusions remain consistent, or does your method have additional advantages?
- The paper is missing citations to key references in the field. Including these references would provide a more comprehensive context and support for the presented work.
- Some typos:
    - Figure 1 is somewhat compressed, and there seem to be spelling errors in 'exstrinstic' and 'instrinstic.'
    - Additionally, there is no space between sentences in line 439.

**Suitability:**

3

---

### Official Review · Reviewer_NVXu · 2024-05-24

**Rating:** 4
**Confidence:** 3

**Summary:**

This paper works on parameter-efficient fine-tuning (PEFT) for medical visual language models (Med-VLM). It discusses PEFT techniques, such as intrinsic PEFT, including fine-tuning LayerNorm, Attention, or FFN, and extrinsic PEFT, including LoRA and prefix-tuning. Challenges for medical VLM fine-tuning are also analyzed. Thorough experiments on in- and out-of-domain VQA and IRG tasks using two VLMs, i.e., MISS and LLaVA-Med, with various PEFT methods, are evaluated. Results suggest that fine-tuning LayerNorm is a promising way for fine-tuning medical VLMs.

**Strengths:**

- The paper works on an interesting topic, PEFT for medical VLM, under the rapid development of medical VLM and their intense training.
- The experiments are extensive. The paper benchmarks two models on many fine-tuning situations, including VQA and text report generation tasks in the in-domain, out-of-domain, and zero-shot settings.
- The experimental results and findings may help in future work on effectively discovering medical VLMs.

**Limitations:**

- Some findings are not evident, as the trend or trade-off is not apparent (subtle difference and unpredictable performance with more trainable parameters). The paper can summarize the main findings in each scenario and guidelines on facing new VLMs on new medical tasks.
- Given some subtle differences in the results, it is better to provide a statistical analysis.
- Besides tables, a plot of performance vs. parameters may be a better way to visualize the result intuitively.
- There is no analysis for Figure 3.

Minors:
- Is the accuracy evaluation robust?
- It is better to include the results for medical VLM without fine-tuning.
- The GPU, memory requirement, and training time should be listed for reference.
- For the related work discussion, it is better to write about the techniques mentioned besides citations for easier reading.
- Figure 2, at the bottom left, should be explained.
- $\phi$ in Eq. 3 is not mentioned.
- The text has some typos, e.g., missing spaces and periods.

**Suitability:**

3

---

### Official Review · Reviewer_ury5 · 2024-05-24

**Rating:** 3
**Confidence:** 2

**Summary:**

This paper focuses on the efficient fine-tuning of medical vision language models. Through systematic experiments, it has been found that LayerNorm can serve as an efficient method for fine-tuning across both large-scale and small-scale Med-VLPs.

**Strengths:**

1. The paper is well organized and written.
2. The paper conducts a large number of systematic experiments and confirms that LayerNorm is useful for fine-tuning the vision language model.

**Limitations:**

However, LayerNorm as an effective fine-tuning method has been studied in cv [2] and nlp [1] fields. The unique role of Layernorm in Med-VLP is not reflected.

Moreover, considering the differences between general vlms and med-vlms, such as scarce data, and short text as shown in section 3.4, the paper does not discuss these specific questions in the medical field in detail. How does Layernorm alleviate the problem of scarce data in the medical field, as an effective fine-tuning method of vision language model?

Besides, figure 2 makes me confused, that there is a CT image in the flowchart, but the question in the right side says "Describe the given chest x-ray image in detail.". Maybe you can provide related pictures related to the problem to make Figure 2 clearer.

Finally, the article mentions MRI and CT on line 287, but does not talk about their related experiments.


[1] Houlsby, Neil, et al. "Parameter-efficient transfer learning for NLP." International conference on machine learning. PMLR, 2019.
[2] Basu, Samyadeep, et al. "Strong Baselines for Parameter-Efficient Few-Shot Fine-Tuning." Proceedings of the AAAI Conference on Artificial Intelligence. Vol. 38. No. 10. 2024.

**Suitability:**

3

---

### Official Review · Reviewer_TvNq · 2024-05-25

**Rating:** 4
**Confidence:** 4

**Summary:**

The work explore different strategies of fine-tuning medical visual language models over popular technique like LoRA. It shows the impact of fine-tuning Layer Normalization (LayerNorm) layers, Feedforward Neural Networks and Attention layers instead adding some components in LoRA on the Med-VLMs. With the validation on 4 VQA and radiology report generation dataset, the study included that ayerNorm fine-tuning demonstrates superior adaptability and scalability.

**Strengths:**

1. The manuscript innovatively investigates the fine-tuning of intrinsic Med-VLM parameters, addressing a significant gap in the field's understanding of efficient adaptation strategies.

2. The study presents a comprehensive empirical evaluation across diverse tasks and model scales, offering valuable insights into the efficacy of intrinsic parameter fine-tuning compared to established PEFT approaches.

3. The research identifies LayerNorm fine-tuning as a promising technique, exhibiting a desirable balance between efficiency and performance, particularly for large-scale Med-VLMs.

**Limitations:**

1. Inconsistent LayerNorm Improvement: The observed performance improvement with LayerNorm fine-tuning is not consistent across all experimental results, raising questions about the generalizability of this approach.

2. Potential for Catastrophic Forgetting: A key goal of PEFT methods like LoRA is to adapt models to new knowledge while retaining previously learned information. Since the proposed strategy focuses on fine-tuning intrinsic model components, it's crucial to assess whether it leads to catastrophic forgetting. This could be investigated by evaluating performance on the original dataset(s) used for pre-training.

3. Limited Comparison with LoRA Variants: The study does not compare LayerNorm fine-tuning with recent LoRA variants like QLoRA, AdaLoRA, or MoRA.  Including these comparisons would provide a more comprehensive understanding of how the proposed method stacks up against other state-of-the-art PEFT techniques.

4. Unexplored Combination with LoRA: It would be valuable to explore the performance of combining LayerNorm fine-tuning with LoRA, as this could potentially lead to further improvements in efficiency and accuracy. An experiment investigating this hybrid approach would be a valuable addition to the study.

**Suitability:**

3

---

### Meta-Review · Area_Chair_wZiR · 2024-07-01

**Recommendation:** Accept (Poster)
**Confidence:** 3

**Metareview:**

The paper explored various strategies of fine-tuning medical visual language models over popular technique like LoRA. It shows the impact of fine-tuning Layer Normalization (LayerNorm) layers, Feedforward Neural Networks and Attention layers instead adding some components in LoRA on the Med-VLMs. With the validation on 4 VQA and radiology report generation dataset, the study included that ayerNorm fine-tuning demonstrates superior adaptability and scalability.

The main conclusion made by the authors is that for medical data with smaller scale and poorer annotation quality, finetuning LN layers is better than other PEFT methods.

After the rebuttal, the paper receives three borderline accept and 1 weak reject. The AC found the authors solved most the reviewers' concerns and the conclusion is somewhat ineresting to share to the community.